# Evolution of Microstructural Characteristics of Carbonated Cement Pastes Subjected to High Temperatures Evaluated by MIP and SEM

**DOI:** 10.3390/ma15176037

**Published:** 2022-09-01

**Authors:** Yongqiang Li, Yaoming Luo, Hangyu Du, Wei Liu, Luping Tang, Feng Xing

**Affiliations:** 1Guangdong Provincial Key Laboratory of Durability for Marine Civil Engineering, Shenzhen University, Shenzhen 518060, China; 2Key Laboratory for Resilient Infrastructures of Coastal Cities (Ministry of Education), College of Civil and Transportation Engineering, Shenzhen University, Shenzhen 518060, China; 3Key Laboratory of Earthquake Engineering and Engineering Vibration, Institute of Engineering Mechanics, China Earthquake Administration, Harbin 150080, China; 4Division of Building Technology, Chalmers University of Technology, 41296 Gothenburg, Sweden

**Keywords:** carbonation, high temperatures, microstructure, SEM, MIP

## Abstract

The microstructural evolutions of both uncarbonated and carbonated cement pastes subjected to various high temperatures (30 °C, 200 °C, 400 °C, 500 °C, 600 °C, 720 °C, and 950 °C) are presented in this study by the means of mercury intrusion porosimetry (MIP) and scanning electron microscopy (SEM). It was found that the thermal stabilities of uncarbonated cement pastes were significantly changed from 400 to 500 °C due to the decomposition of portlandite at this temperature range. More large pores and microcracks were generated from 600 to 720 °C, with the depolymerization of C-S-H. After carbonation, the microstructures of carbonated cement pastes remained unchanged below 500 °C and started to degrade at 600 °C, due to the decompositions of calcium carbonates and calcium modified silica gel. At 950 °C, both uncarbonated and carbonated cement pastes showed a loosely honeycombed microstructure, composed mainly of β-C_2_S and lime. It can be concluded that carbonation improves the high-temperature resistance of cement pastes up to 500 °C, but this advantage is lost at temperatures over 600 °C.

## 1. Introduction

Facing the urgent necessity to deal with global warming and climate change, many organizations and governments have declared their ambitions to reach carbon neutrality in 2050 [1]. To achieve this goal, carbon capture and storage technologies are seen as potential ways to reduce the emissions of greenhouse gases (mainly CO_2_) [2], especially for cementitious materials, which can capture and store the CO_2_ safely and permanently [3,4,5,6]. The carbonation reaction happens when the main hydration products in the hydrated cement, i.e., portlandite (CH) and calcium silicate hydrate (C-S-H), react with the penetrated CO_2_ to generate calcium carbonate (CaCO_3_) and calcium modified silica gel (Equations (1) and (2)) [7,8]. Other unhydrated products, including tricalcium silicate (C_3_S), dicalcium silicate (C_2_S), and ettringite, may also be consumed during carbonation [9,10]. Carbonation may cause the neutralization of concrete to trigger the corrosion of steel reinforcements [11] and accelerate the transport rate of some corrosive ions, such as chloride ions [12], while the pore structure after carbonation is refined at the same time to form a denser microstructure with higher strength [13]. For the concrete exposed to the atmospheric environment, the carbonation process usually cannot be ignored, considering its long service life (more than several decades) and continuous penetration of CO_2_ gas. Particularly for concrete in subsea tunnels, or some other underground constructions, the carbonation depth is more noticeable due to the high CO_2_ concentration in the environment [14].
(1)$$\mathrm{Ca}{\left(\mathrm{OH}\right)}_{2}+{\mathrm{CO}}_{2}\to {\mathrm{CaCO}}_{3}+H_{2}O$$
(2)$${\left(\mathrm{CaO}\right)}_{x}\left({\mathrm{SiO}}_{2}\right){\left(H_{2}O\right)}_{z}+{\mathrm{yCO}}_{2}\to {\mathrm{yCaCO}}_{3}+{\left(\mathrm{CaO}\right)}_{x-y}{\mathrm{SiO}}_{2}{\left(H_{2}O\right)}_{t}+\left(z-t\right)H_{2}O$$

As one of the most critical challenges to all the cement-based materials, the durability performance of concrete at high temperatures should be emphasized, considering the countless fire disasters which have happened in the past [15]. This is not only related to the personnel security during an accidental fire, but also determines the post-fire repairing strategies. Numerous studies have been performed on the high-temperature performance of cement-based materials regarding their macro and micro properties [16,17,18,19], while the degradation performance of carbonated cement-based materials is rarely seen in the literature [20,21]. It has been reported that the decompositions of CH and C-S-H should be responsible for the degradation of uncarbonated concrete subjected to the high temperature [22,23]. The decompositions of both CH and C-S-H can increase the porosity, making the microstructure more porous, and finally, resulting in the collapse of microstructure under ultrahigh temperatures. However, for the carbonated concrete, most of the CH and C-S-H have been consumed after carbonation. The main phases in carbonated pastes are CaCO_3_ and calcium modified silica gel. The precipitated CaCO_3_ shows a higher decomposition temperature than the CH [24], suggesting a higher anti-fire performance of carbonated cement-based materials, while the decomposition temperature of calcium modified silica gel is found to be primarily above 500 °C [25]. Therefore, the ability of carbonated concrete to resist high temperatures marks a distinct difference between carbonated and uncarbonated concrete. For the concrete with remarkable carbonation depth, the exteriorly carbonated concrete is the first area to be heated, if an accidental fire occurs. It is meaningful to study the microstructural performances of carbonated concrete at high temperatures for a better estimation of the durability performance of fired concrete, especially in some regions where the universal carbonation depth is significant [26]. The present research work is a continuation of our previous studies [25,27], which demonstrated the chemical and mineralogical changes of carbonated cement pastes subjected to different high temperatures up to 950 °C. In this work, the microstructural evolutions of carbonated cement pastes subjected to various high temperatures were further investigated by mercury intrusion porosimetry (MIP) and scanning electron microscopy (SEM). The effects of carbonation on the fire resistance of cement pastes were evaluated by comparing the microstructural results of uncarbonated and carbonated samples under different high temperatures. It is anticipated that findings from this research work could provide some theoretical support for the post-fire assessment and repair of fired concrete from the point view of cement pastes.

## 2. Experimental Procedures

### 2.1. Sample Preparation

Cement pastes, with a water to cement ratio (w/c) of 0.56, were poured into 20 mm diameter and 100 mm height cylindrical plastic tubes and then sealed, after sufficient mixing. The samples were then rotated at a speed of 10 rpm/min for 24 h to promote homogeneity and prevent segregation. These tubes were then cured under the sealed condition at 22 ± 2 °C for another 27 days. After demolding, the cylinders were cut to slices, with a thickness of around 3 mm, using a diamond cutting saw for easier carbonation. The slices were put into an accelerated carbonation chamber with 20% CO_2_ [28], and the mass variations were recorded during the carbonation process. It was found that sample mass was stabilized after 100 days of carbonation, and all fractured surfaces appeared colorless after spraying with phenolphthalein solution. It is therefore determined that 100 days of carbonation accounted for the complete carbonation. At the same time, a portion of the cement pastes were kept sealed in the tubes as uncarbonated samples for future comparison. Next, to compare the results at room temperature (around 30 °C), both carbonated and uncarbonated pastes were simultaneously heated to various high temperatures: 200 °C, 400 °C, 500 °C, 600 °C, 720 °C, and 950 °C, with a heating rate of 10 °C/min and maintained at the specified temperature for 90 min. The visual appearances of the carbonated samples under different high temperatures were recorded. Finally, the samples were naturally cooled down in the oven to the room temperature, removed, and stored in a vacuum desiccator prior to experimental measurement. The schematic diagram of the experimental process is shown in Figure 1. It is assumed that properties at the edge or center of the cylindrical samples are the same, i.e., having the same hydration or carbonation degree, so that microstructural results from any part of sample could reveal the real situation under different high temperatures, and thus the samples used for microstructural characterization were randomly chosen. 

### 2.2. Mercury Intrusion Porosimetry (MIP)

Small pieces of samples (around 1 g) were collected and dried in a vacuum oven at 50 °C for 3 days before MIP measurement. MIP test was carried out using an Autopore IV 9500 instrument, with a theoretically minimum measured pore size of 5 nm. The relationship between the pore radius (r) and the pore pressure (p) is expressed as Equation (3) [29].
(3)$$p=-\frac{2\gamma cos\left(\theta \right)}{r}$$
where γ is the surface tension of mercury (0.485 N/m) and θ is the contact angle between mercury and the pore surface (130°).

### 2.3. Scanning Electron Microscopy (SEM)

Thin pieces from the same batch were fractured for SEM observation with ZEISS Gemini equipment at the secondary electron mode. The samples were sputtered with a thin layer of gold before observation. In addition, an energy dispersive spectroscopy (EDS) system was equipped to detect the elemental compositions in the areas of interest.

## 3. Results and Discussions

### 3.1. Evolutions of Appearances

The evolutions in the appearance of the carbonated samples under different high temperatures are shown in Figure 2. It can be seen that the visual appearance of the carbonated samples was unchanged below 500 °C, indicating an integrated paste structure within this temperature. Once the temperature was elevated to 600 °C, a crack appeared on the surface, and more cracks were generated with further increases in temperature at to 720 °C and 950 °C, which is related to the microstructural deterioration, as shown below.

### 3.2. MIP Results

The porosity changes for both uncarbonated and carbonated cement pastes subjected to different high temperatures are shown in Figure 3. The porosity changes of cement pastes under high temperatures could be attributed to the decomposition of hydration or carbonation products and the thermal cracks generated due to thermal expansion. For the uncarbonated cement pastes, the porosity slowly increased below 400 °C due to the complete decomposition of ettringite below 200 °C and the partial decomposition of CH at 400 °C [30]. A significant step occurred between 400 °C and 500 °C (from 40.1% to 53.8%) due to the complete dehydroxylation of CH to lime, creating additional capillary porosity [31]. When the temperature was higher than 600 °C, the porosity increased almost linearly with the increase in temperature. This is mainly caused by the depolymerization of the C-S-H phases to crystal β-C_2_S [19,24,32,33]. However, for the carbonated cement pastes, the porosity under different high temperatures was always lower than for the corresponding uncarbonated samples. Below 400 °C, the porosity was almost unchanged. The increase in porosity for the carbonated samples started from 500 °C, which is attributed to the decompositions of both crystal and amorphous CaCO_3_ [34]. Subsequently, the porosity continuously increased with higher temperatures, due to the further decomposition of CaCO_3_ polymorphs (including calcite, vaterite, and aragonite) and calcium modified silica gel [25]. The porosity changes in carbonated pastes under high temperatures roughly obey the exponential function.

Apart from the total porosity, the pore size distribution plays a crucial role in the durability performance of concrete, as shown by the normalized compositions in Figure 4. Based on the theory provided by Wu et al. [35], four categories of pores can be identified according pore size: harmless pores (<20 nm), less harmful pores (20–50 nm), harmful pores (50–200 nm), and more harmful pores (>200 nm), as the extent of harm is determined by the concrete strength and permeation. For the uncarbonated cement pastes at 30 °C, the pore size is mainly distributed in the range of <50 nm. From 30 to 200 °C, a few more harmful pores, with pore sizes 50–200 nm, were generated. Later, above 400 °C, many more harmful pores, which are usually seen as the medium to promote the transport abilities of carbon dioxide and chloride ions, damaging the durability performance of concrete [36], were generated in the cement matrix due to consumption of CH, and in this case, the mechanical performance of the paste could be expected to decrease above 400 °C. In addition, when subjected to 950 °C, the more harmful pores were almost exclusively generated, which should more accurately be described as cracks, indicating a total collapse of the microstructure under exposure to a 950 °C temperature, which is consistent with the visual observation that after exposure to this temperature, the remaining structure was very loose and easily collapsed to powder under slight compression.

Compared with the uncarbonated samples, carbonation refined the microstructure by blocking the larger pores with the precipitated calcium carbonates [37]. From 30 °C to 400 °C, the cumulative pore size distribution was nearly the same, confirming the stability of the carbonated cement matrix below 400 °C. That is, the CaCO_3_ and calcium modified silica gel are not decomposed under 400 °C. However, it is interesting to note that the percentages of pores with size >200 nm decreased from 30 to 400 °C, but increased above 500 °C (Figure 4). A probable explanation is that CaCO_3_ was transformed from an amorphous to a crystal state under 400 °C to refine the pore structure [27], but above 500 °C, both the crystal CaCO_3_ and calcium modified silica gel were decomposed, leading to the occurrence of voids, generating the more harmful pores. For the carbonated cement pastes in this study, the main composition is CaCO_3_, with a mass percentage of about 55% [25]. After the decomposition of CaCO_3_ under high temperatures, the mechanical properties of carbonated cement pastes can be predicted to decline after 500 °C. Especially in the conditions over 600 °C, the pore size distribution of uncarbonated and carbonated samples was quite similar, suggesting the similar deteriorations of uncarbonated and carbonated cement pastes over 600 °C. Hence, from the MIP perspective, the carbonated cement pastes are more stable than the uncarbonated pastes under 400 °C, but presents similar degradation above 500 °C.

However, it was noted that the cement pastes became quite weak after treatment with high temperatures above 720 °C. Even gentle pressure may lead to the collapse of the whole matrix. When the MIP test is performed, the mercury pressure may aggravate the deterioration of the sample, making it possible to obtain results that may deviate slightly from the actual situation. The more harmful pores and cracks, as typically seen in the results of 950 °C in Figure 4, are also difficult to distinguish using only MIP results. Another drawback of the MIP method is the ink-bottle effect which occurs during the mercury intrusion [30,38]. MIP measures only the pore entry sizes and not the real pore sizes of the sample, which could result in the overestimation of smaller pores. Unfortunately, this adverse effect cannot be overcome due to the limitations of this measurement method. These drawbacks regarding the MIP results should be considered, and thus SEM observation has been supplemented to obtain a better understanding of the microstructural changes under high temperatures.

### 3.3. SEM Results

The micro-morphologies of uncarbonated cement pastes with elevated temperatures are shown in Figure 5, with the magnification of some areas of interest. After being subjected to 200 °C, although the decomposition of ettringite was completed at this temperature [39], the micro-morphologies showed no obvious change, due to the intact mixtures of CH and C-S-H, proving the integrity of pastes under 200 °C. At 400 °C, part of CH was decomposed, with several cracks showing on the surface of the CH crystals, which is consistent with the MIP result that the percentage of the more harmful pores has a sharp increase at this temperature. At 500 °C, no CH crystal remained, but only aggregations of granules were found in the SEM images, which according to previous studies [39] and EDS results for spot 1, should correspond with the morphology of lime. Several microcracks of broad widths were also found in the image, further coarsening the pores due to the shrinkage by 33% volume during the CH decomposition to lime [40]. When the temperature reached 600 °C, the microcracks propagated throughout the whole matrix, with the maximum width of more than 1 μm. Although our previous research showed that C-S-H was severely depolymerized at 600 °C [32], the fibrous morphology of C-S-H was still visible. At 750 °C and 950 °C, none of C-S-H morphology could be seen in the SEM images. The whole cement matrix was destroyed by heat treatment to form a loosely honeycombed structure. Spherical particles with diameters around 20 μm were found throughout the images. EDS results showed that the Ca/Si ratio at spot 2 was 2.27, suggesting that the spherical particles are mainly composed of β-C_2_S and lime, but minor proportions of C_3_S may exist as well.

Figure 6 presents the microstructures of carbonated cement pastes heated under various temperatures. In contrast to the uncarbonated sample, the morphology at room temperature was obviously denser, owing to the precipitation of CaCO_3_ after carbonation. However, the morphology of crystalline CaCO_3_ cannot be seen in the SEM image. A suspicious position of pure CaCO_3_ (spot 3) showed a high content of element C and the signal of Si in the EDS results, suggesting that the compounds grown in the pore wall could be mixtures of calcium carbonates, calcium modified silica gel, decalcified C-S-H, and uncarbonated C-S-H [41]. With the temperature being elevated from 30 °C to 500 °C, the microstructures were still closely connected, which is consistent with the MIP results that pore structures in this temperature range were not dramatically changed, although partial decomposition of CaCO_3_ occurs at 500 °C [25]. However, the cement pastes became poriferous when the temperature reached at 600 °C. Many large pores could be found in the images, leading to an unstable microstructure. After the exposure to a higher temperature, at 720 °C, the microstructure further collapsed, with the appearances of significant cracks in widths of about 2 µm. Finally, at 950 °C, the intact matrix was replaced by aggregations of circular particles to form the most porous microstructures, in which the binding capacity between each particle became very weak. The elemental compositions of the circular particles (spot 4) were close to those of the uncarbonated sample at the same temperature (spot 2), with Ca/Si at 2.52. Considering that the formation temperature of the C_3_S was much higher than 950 °C [42], the main compositions at 950 °C should be the mixtures of crystalline C_2_S and lime. In general, the SEM results were in line with the MIP results. The coarsening tendency and porosity occurring when the samples were subjected to high temperatures was clearly shown by the SEM images, and the more harmful pores and cracks that are difficult distinguished by MIP can be seen in the SEM results. It should be pointed out that thermal expansion may also lead to the generation of cracks, which, nevertheless, is not mentioned in this study due to the lack of results regarding this effect. 

By combining MIP and SEM results, we could see that the degradation process of cement pastes begins with the generation of microcracks, or large holes. For the uncarbonated cement pastes, the integrity of the pastes is maintained by the hydration products (mainly CH and C-S-H). Under 400 °C, the water that is weakly linked to the hydration products, especially the C-S-H, is evaporated with the decomposition of the hydration products, while above 500 °C, the bound water in CH and C-S-H is released [43]. Once the hydration products are decomposed, numerous vacancies and defects appear, accompanied by volume shrinkage, contributing to weaker inner strength and easily initiating and propagating cracks [44]. In addition, the evaporation of water can generate high tensile stresses in the pore walls, leading to the expansion of cracks or even explosive spalling [18]. The cement paste made from ordinary Portland cement may, therefore, tolerate up to 400 °C, when subjected to high temperatures.

On the contrary, for the carbonated cement pastes, most of the calcium ions were precipitated to CaCO_3_. The integrity of the paste is controlled by the stability of CaCO_3_, as well as by the calcium modified silica gel. Shrinkage will start with the decomposition of CaCO_3_ to lime, followed by the evaporation of water from the depolymerization of calcium modified silica gel [40]. During these processes, the microcracks occur and propagate with the rising temperatures, creating a porous microstructure after heat treatment. From above results, it can be seen that the microstructures of carbonated cement pastes remain unbroken up to 500 °C. Thus, the anti-fire performance of carbonated pastes increases to 500 °C, as opposed to 400 °C for the uncarbonated pastes. The mechanical results from Meng et al. [20] also verified the positive effect of carbonation on enhancing the high temperature performance of cement pastes. Moreover, it is also noted that both uncarbonated and carbonated cement pastes show loose microstructures with low integrity over 600 °C, in which the porosity is >45% in MIP results, suggesting severe deterioration over 600 °C and the disappearance of the superiority of carbonation above this temperature. The anti-fire performance of both uncarbonated and carbonated cement pastes should be considered to be the same at temperatures above 600 °C.

## 4. Conclusions

The aim of this investigation is to study the effect of carbonation on the microstructural characterizations of cement pastes subjected to different high temperatures. Several conclusions can be drawn, based on the experimental results.

For the uncarbonated cement pastes, the microstructures are significantly changed at temperatures from 400 to 500 °C, due to the decomposition of CH. With further elevation of temperature from 500 to 720 °C, more harmful pores and microcracks with broader width and longer length are generated by the depolymerization of C-S-H. At 950 °C, only a loosely honeycombed microstructure remains.For the carbonated cement pastes, there is no significant change in microstructure at temperatures ranging from 20 to 400 °C. Over 500 °C, the pastes are severely damaged by the elevation in temperature due to the decomposition of CaCO_3_ and calcium modified silica gel. At 950 °C, only the more harmful pores and loose microstructures, mainly composed of β-C_2_S and lime, remain in the sample.Carbonation can provide better high-temperature resistance under 500 °C due to the higher decomposition temperature of calcium carbonate. However, over 600 °C, the anti-fire performance of both uncarbonated and carbonated cement pastes should be considered to be the same.

## Figures and Tables

**Figure 1 materials-15-06037-f001:**
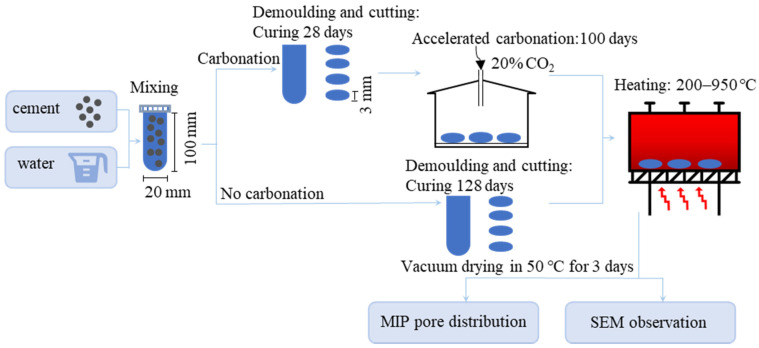
Schematic diagram of the experimental process.

**Figure 2 materials-15-06037-f002:**
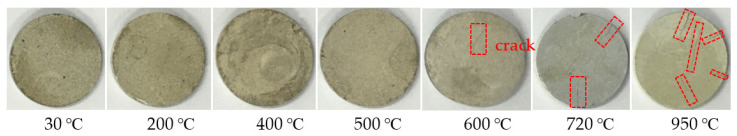
Evolutions in appearance of carbonated samples under high temperatures.

**Figure 3 materials-15-06037-f003:**
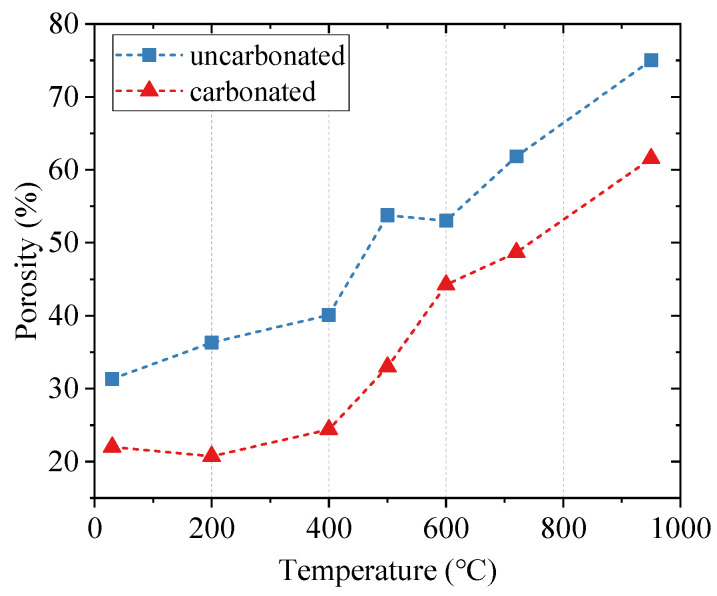
Porosity changes in cement pastes under elevated temperatures.

**Figure 4 materials-15-06037-f004:**
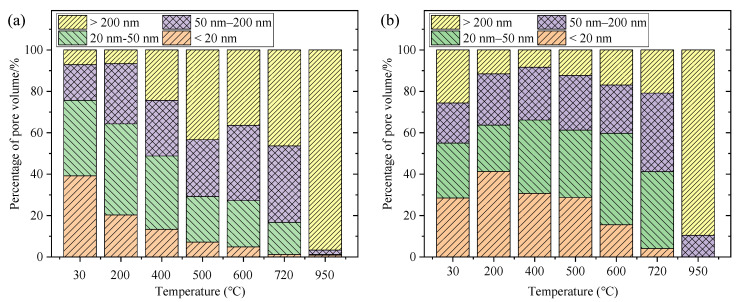
Compositions percentages of uncarbonated (**a**) and carbonated (**b**) cement pastes under different high temperatures.

**Figure 5 materials-15-06037-f005:**
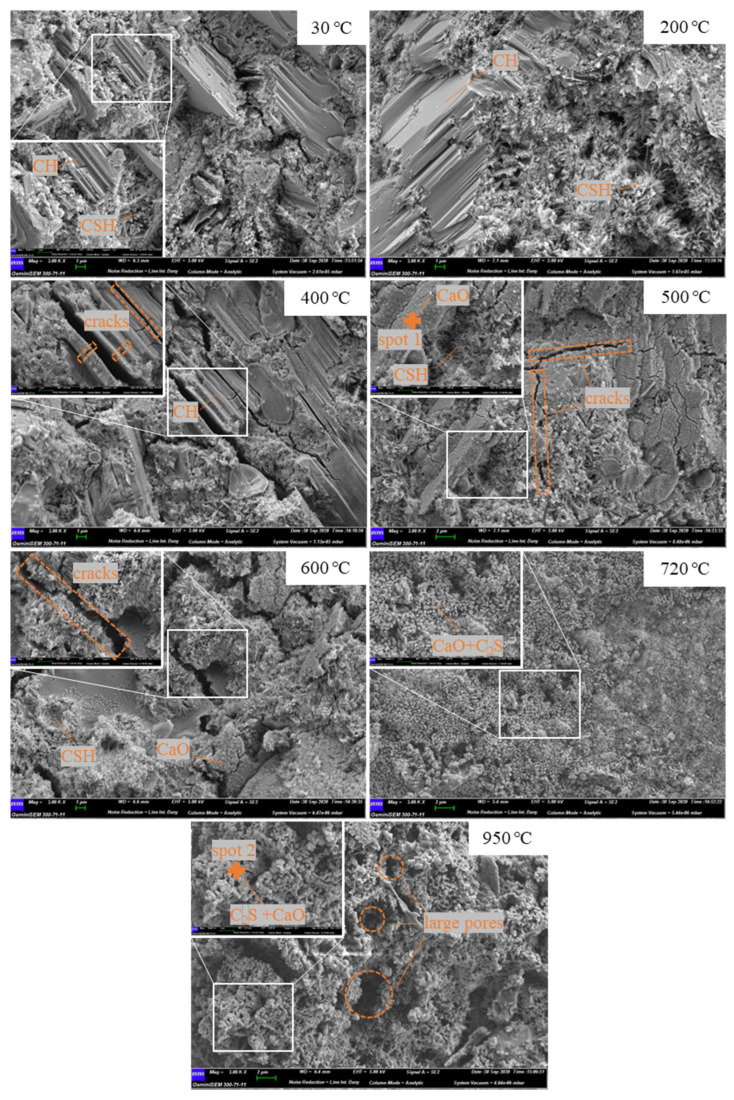
SEM images of uncarbonated cement pastes subjected to different high temperatures.

**Figure 6 materials-15-06037-f006:**
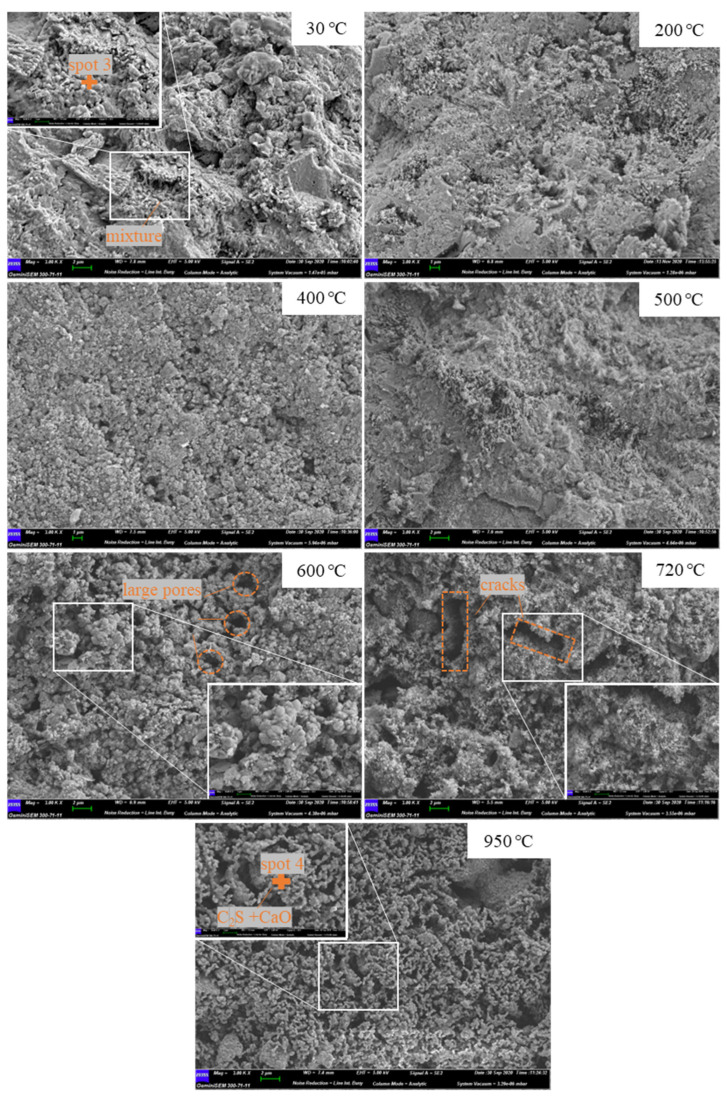
SEM images of carbonated cement pastes subjected to different high temperatures.

## Data Availability

The data is available after request.

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
