# Peer review of "Evolution of Microstructural Characteristics of Carbonated Cement Pastes Subjected to High Temperatures Evaluated by MIP and SEM"

_materials, 2022, doi:10.3390/ma15176037_

Round 1

Reviewer 1 Report

Abstract should inform to the reader what is the logic behind this work, its written as a short communication and only if readers know previous work of authors, they can understand it.

Line 30-32: What exactly authors mean from this ?

Line 61: Which previous work, do cite that here.

Line 68: concrete and cement paste behave differently in thermal condition, how to map these two?

Line 72: How mixing in this small tube is done and how authors assume its perfect.

Line 74: Based on what code provision this is designed?

Fig 3: How? How the pores change due to temperature, finding is good, but how ?

Line 136: How is this classification done, harmful means what? How harmful is decided?

Line 139: Here harmful is decided with pore size?

Line 164: How its concluded that the paste become so weak?

References are not in format; many are in different format.

References are not updated one, some are really obsolete, think, if its mandatory leave, or else change

Fig 7,8,9 interpretation is not sufficient, these papers major work is in these three figures, interpretation should be good, scientific and acceptable.

Line 248, 253: Why?

Is the temperature increase and pore structure change due to thermal impact or chemical modification?

What is the necessity to study upto 950 degrees? Where in field this temperature exists?

Third point of conclusion nullify the logic of this paper, please check, section 3.3 seems to be vague.

Author Response

Q1: Abstract should inform to the reader what is the logic behind this work, its written as a short communication and only if readers know previous work of authors, they can understand it.

Answer 1: We are sorry for the confusing logic in last version of manuscript. The Abstract has been improved to our best in revised manuscript. Please kindly see details in revised manuscript.

Q2: Line 30-32: What exactly authors mean from this?

Answer 2: Sorry for the misunderstanding. The sentence has been modified as “To achieve this goal, carbon capture and storage technologies are seen as potential ways to reduce the emissions of greenhouse gas (mainly the CO2) [2], especially for the cementitious materials which can capture and store the CO2 safely and permanently [3-6]” in Lines 28-31 of revised manuscript.

Q3: Line 61: Which previous work, do cite that here.

Answer 3: Thank you for your comment. The previous work has been cited in Line 63 of revised manuscript.

Q4: Line 68: concrete and cement paste behave differently in thermal condition, how to map these two?

Answer 4: The different behaviors of concrete and cement paste in thermal condition should be ascribed to the existence of aggregates in concrete. The compositions of aggregates, including quartz and limestone, can vary in different regions, and the decomposition temperatures of aggregates and hydration/ carbonation products deviate a lot. However, the influence of aggregates on the thermal performance of concrete is not addressed in this study, and the findings from this research work could provide some theoretical supports for post-fire assessment and repairment of fired concrete from the point view of cement pastes as mentioned in Line 70 of revised manuscript.

Q5: Line 72: How mixing in this small tube is done and how authors assume its perfect.

Answer 5: Cement pastes with water to cement ratio (w/c) of 0.56 were poured into the sealed cylindrical plastic tubes with 20 mm diameter and 100 mm height after sufficient mixing and rotated with the speed of 10 rpm/ min in 24 hours to promote its homogeneity and prevent segregation. These tubes were then cured under the sealed condition at 22±2 ℃ for another 27 days. Based on the mixing procedure mentioned in Lines 73-76 of revised manuscript, the prepared sample is assumed to be homogenous enough.

Q6: Line 74: Based on what code provision this is designed?

Answer 6: The slices were put into accelerated carbonation chamber with 20 % CO2 [27], and the mass variations were recorded during the carbonation process. It was found that sample mass has been stabilized after 100 days’ carbonation, and all fractured surfaces showed colorless by spraying with phenolphthalein solution. It is therefore that the state after 100 days’ carbonation was accounted as the complete carbonation. The above explanation has been added in Lines 77-81 of revised manuscript.

Q7: Fig 3: How? How the pores change due to temperature, finding is good, but how?

Answer 7: The porosity changes of cement pastes under high temperatures could be contributed to the decompositions of hydration or carbonation products and the generated thermal cracks due to thermal expansion. The above explanation has been added in Lines 119-121 of revised manuscript to explain the reason leading to the porosity change under high temperatures. Besides, the empirical equations in Fig. 3 have been deleted in revised manuscript to eliminate the potential misinterpretation.

Q8: Line 136: How is this classification done, harmful means what? How harmful is decided?

Answer 8: Based on the theory provided by Wu et al. [33], four categories of pores can be identified according to its pore size: harmless pores (< 20 nm), less harmful pores (20-50 nm), harmful pores (50-200 nm) and more harmful pores (>200 nm), as the extent of harm is decided by the concrete strength and permeation. The above explanation has been added in Lines 137-140 of revised manuscript.

Q9: Line 139: Here harmful is decided with pore size?

Answer 9: Yes. A little more harmful pores, with pore size 50-200 nm, were generated. The above explanation has been added in Line 142 of revised manuscript.

Q10: Line 164: How its concluded that the paste become so weak?

Answer 10: However, it was noted that the cement pastes have become quite weak after the treatment of high temperature above 720 ℃. Even soft touching may lead to the collapse of the whole matrix. The above explanation has been added in Lines 166-167 of revised manuscript.

Q11: References are not in format; many are in different format.

Answer 11: The formats of references have been carefully checked before resubmission. Please kindly see the References in revised manuscript.

Q12: References are not updated one, some are really obsolete, think, if its mandatory leave, or else change

Answer 12: Thank you for your suggestion. The obsolete reference [9] has been updated, and other old and unnecessary references, including [16, 38, 43] in the first version of manuscript, have been deleted. In revised manuscript, most of the references are published after 2010.

Q13: Fig 7,8,9 interpretation is not sufficient, these papers major work is in these three figures, interpretation should be good, scientific and acceptable.

Answer 13: Thanks for review’s good comments. Many efforts have been performed to deeply analyze the results of SEM, and the Fig. 9 has been deleted for better demonstration. Please kindly see the details in revised manuscript.

Q14: Line 248, 253: Why?

Answer 14: The first two conclusions were given based on the experimental results as the decompositions of portlandite and calcium carbonate with the increasing temperatures were obtained from MIP and SEM images. Please see details in Conclusion of revised manuscript.

Q15: Is the temperature increase and pore structure change due to thermal impact or chemical modification?

Answer 15: The changes of pore structures should be related to the chemical modifications caused by the decompositions of hydrated or carbonated phases under high temperatures. It is true that thermal expansion may happen at high temperature as well to induce the pore structure change, but this factor is not considered due to lacked information regarding this effect. 

Q16: What is the necessity to study up to 950 degrees? Where in field this temperature exists?

Answer 16: The temperature in an actual fire site can reach above 900 ℃ [1]. The situation of 950 ℃ was chosen in this study to investigate the highest temperature in a real fire condition.

[1] B. Fernandes, A. Gil, F. Bolina, et al. Microstructure of concrete subjected to elevated temperatures: Physico-chemical changes and analysis techniques, Rev. IBRACON Estrut. Mater,, 10 (2017), 838-863.

Q17: Third point of conclusion nullify the logic of this paper, please check, section 3.3 seems to be vague.

Answer 17: The carbonation can provide better high-temperature resistance under 500 ℃ due to the higher decomposition temperature of calcium carbonate. While over 600 ℃, the anti-fire performance of both uncarbonated and carbonated cement pastes should be considered as the same. The above mentioned third point of conclusion has been corrected in Lines 256-258 of revised manuscript. In addition, the section 3.3 has been carefully improved to avoid potential unclear. Please kindly see details in revised manuscript.

Reviewer 2 Report

Row 82: Do you check the properties at the edges and the center?

Row 106: improvrmrnts of temperatures? Increase of temperature???

Row 118, 119: The depolymerization of C-S_H-phases is topic of a lot of publication. You cite only one.

Row 133, fig. 3: The differences between uncarbonated and carbonated material are clear. The empirical equations are not necessary, complete superflous, because without chemical or physical background. Either you can give an background for the s-shaped curve or you cancel the equations.

Fig. 3, 4, 5, 6: 4 figures on MIP. I think fig. 3 and fig. 6 are enough. Cancel fig. 4 and 5.

Fig. 7 and 8 must be placed in the text directly. For the reader it was more easy to replace the signs (a), (b) etc. by the real temperatures. 

Fig. 9: Cancel

Author Response

Q1: Row 82: Do you check the properties at the edges and the center?

Answer 1: Yes, the properties at the edges and the center have been visually checked by spraying with the phenolphthalein solution to guarantee all samples have been carbonated before microstructural measurements. Besides, the samples have been rotated for 24 hours after casting as mentioned in Line 75 of revised manuscript to promote its homogeneity. It is therefore the properties at the edges and the center are assumed to be the same.

Q2: Row 106: improvements of temperatures? Increase of temperature???

Answer 2: Sorry for the misunderstanding, the expression has been modified as increase of temperature as seen in Line 112 of revised manuscript.

Q3: Row 118, 119: The depolymerization of C-S-H phases is topic of a lot of publication. You cite only one.

Answer 3: Thanks for your good suggestion. Another published references of [19, 24, 33, 34] are cited to prove the depolymerization of C-S-H to β-C2S, as seen in Line 127 of revised manuscript.

Q4: Row 133, fig. 3: The differences between uncarbonated and carbonated material are clear. The empirical equations are not necessary, complete superflous, because without chemical or physical background. Either you can give a background for the s-shaped curve or you cancel the equations.

Answer 4: Thanks for your valuable comments. The s-shaped curve as well as the fitted equations have been deleted in revised manuscript. Please kindly see details in Fig. 3 of revised manuscript.

Q5: Fig. 3, 4, 5, 6: 4 figures on MIP. I think fig. 3 and fig. 6 are enough. Cancel fig. 4 and 5.

Answer 5: The Figs. 4 and 5 have deleted in revised manuscript according to review’s comment.

Q6: Fig. 7 and 8 must be placed in the text directly. For the reader it was more easy to replace the signs (a), (b) etc. by the real temperatures. 

Answer 6: The descriptions of Figs 7 and 8 have been modified according to review’s comment. Please see Figs. 5 and 6 in revised manuscript.

Q7: Fig. 9: Cancel.

Answer 7: The Fig. 9 has been deleted in revised manuscript.